# Experimental Testing and Numerical Bite Simulation of Complete Acrylic Dentures in Eugnathic and Progenic Occlusal Relationships

**DOI:** 10.3390/ma18112427

**Published:** 2025-05-22

**Authors:** Martin Pavlin, Robert Ćelić, Nenad Gubeljak, Jožef Predan

**Affiliations:** 1Tindens Dental Clinic, 2000 Maribor, Slovenia; martin.pavlin@amis.net; 2Department of Removable Prosthodontics, School of Dentistry, University of Zagreb, 10 000 Zagreb, Croatia; celic@sfzg.hr; 3Faculty of Mechanical Engineering, University of Maribor, 2000 Maribor, Slovenia; jozef.predan@um.si

**Keywords:** bite force, finite element method, dental biomechanics, numerical simulations, complete acrylic dentures, occlusal force distribution, occlusal pressure distribution

## Abstract

Complete dentures are exposed to complex masticatory forces that may lead to material fatigue and eventual structural failure. Occlusal relationships, such as eugnathic and progenic, influence the distribution of these forces significantly. Understanding their biomechanical impact is essential for improving denture design and longevity. The aim of this study was to evaluate the mechanical behaviour of complete dentures under bite loads in eugnathic and progenic occlusal relationships, using both experimental testing and numerical simulations. The focus was placed on identifying the conditions that lead to initial damage and the patterns of stress distribution. The material properties of the denture base and artificial teeth were determined through experimental tensile and compressive testing on cylindrical PMMA specimens. The denture geometry was acquired via 3D tomography based on impressions of an edentulous patient. Experimental testing of the denture bite was conducted to determine the force thresholds at which the initial cracks occur. Numerical simulations were carried out using finite element analysis at bite loads of 100 N and 200 N in both occlusal types, incorporating the obtained material parameters. The experimental results showed that the first signs of denture damage occurred at 6400 N in eugnathic occlusion and 7010 N in progenic occlusion. The numerical simulations confirmed that, during occlusion, the pressure is redistributed across multiple contact points, with a broader distribution reducing the localised stress. This redistribution was more efficient in eugnathic occlusion, which reduced the risk of longitudinal cracking in acrylic teeth. In contrast, progenic occlusion showed higher susceptibility to fractures within the acrylic denture base, particularly between adjacent teeth. Both the experimental and numerical approaches demonstrated that occlusal relationships affect the mechanical resilience of complete dentures directly. The findings highlight that eugnathic occlusion offers biomechanical advantages in stress distribution, potentially reducing the risk of fracture. Incorporating occlusal analysis into denture design protocols can enhance clinical outcomes and improve prosthetic longevity.

## 1. Introduction

Partial or complete edentulism affects masticatory efficiency, speech articulation, and facial aesthetics adversely; the fabrication of partial or complete removable dentures serves as a rehabilitative intervention that restores oral function, phonetic clarity, and a natural facial profile, thereby improving the patient’s quality of life significantly. Patients come for an examination twice a year. After two years of using complete dentures, we find that patients’ used dentures show damage on the surface of the upper and lower acrylic teeth. Less often than damage, cracks or breakage of the acrylic base of the upper prosthesis appear. The lower denture cracks less often. Patients report that when a crack occurs, its spread is relatively fast and within two to three days. There can be a complete dysfunction or fracture of the upper denture. Damage to the occlusal surface of acrylic teeth occurs more often in patients with an improper progenic bite when biting hard food and due to lifestyle habits.

Numerous studies, such as that by Campbell et al. [1], show that the number of patients with partial and full dentures increases steadily with age. The lifespan of dentures is also extended with the constant improvement in dental hygiene and the hygiene of complete dentures. As the lifespan of complete dentures increases, so does the likelihood of mechanical damage and the formation of cracks that can lead to fractures. It is common practice to fabricate complete dentures from polymers such as polymethyl methacrylate (PMMA), as described by Narva et al. [2].

Damage often occurs at the point of stress concentration on the upper full denture between the acrylic teeth and the denture base, as shown in Figure 1a.

Figure 1b shows that the formation and growth of the fatigue crack took place practically in concentric circles, the contour of which corresponds to a force at the same time as the penetration. Extensive studies have already been carried out to determine the bite force, e.g., by Laner B. Rosa et al. [3], which show that the range of maximum bite forces in people with healthy teeth can be 500 N to 800 N, while these forces in full dentures are 100 N to 200 N [3,4]. The maximum and average forces were measured with dynamometers in the area of the first molars [3] and also in the last molars [4]. To date, numerous studies have been conducted to analyse the behaviour of dental materials [5,6]. Studies have also been conducted to evaluate the suitability of materials for partial or full dentures [1], as well as numerical simulations using the finite element method [7]. Previous numerical simulations showed the state of displacement and pressure at a certain finite bite force.

The aim of our work was to observe the pressure changes at two bite force levels of complete dentures in two occlusal relationships—eugnathic and progenic—and to monitor the pressure changes on acrylic teeth and denture bases for the two bite forces of 100 N and 200 N mentioned above.

A proper bite is a eugnathic relationship. In this relationship, the upper teeth, the incisors, cover the lower incisors and slide along the surface of the front lower teeth until the premolars and molars come into contact with each other. The molars are the teeth for chewing food.

A progenic relationship is the contact of the lower and upper incisors and occurs when the lower jaw moves forward so that there is no overlap, but rather, there is the first contact of the tips of the upper and lower teeth. Because of the contact of the front incisors, the molars do not fit completely, and food is not chewed completely, which affects digestion negatively.

The aim of this work was to simulate the bite numerically and determine the equivalent stress in the contact of the acrylic teeth in the occlusal relationship of complete dentures, as well as the difference in equivalent stress between the eugnathic and progenic relationship of complete dentures. The results can also be used indirectly to investigate the pressure relationships between the dental contact surfaces of the patient in question, as the complete dentures were fabricated using the patient’s bite impression.

Acrylic teeth and the denture base, as the primary load-bearing components of complete dentures, are subjected to various functional stresses, including compression, tension, shear, and torsion, which may lead to fracture. Additionally, mechanical factors such as variable magnitudes and directions of masticatory loading, differences in the viscoelastic properties and thickness of the oral mucosa, the absence of a precise axis of symmetry, non-uniform denture base thickness, and anatomical variations in the edentulous alveolar ridges of the maxilla and mandible complicate the clinical study of this topic. These factors also influence the distribution and behaviour of the internal stresses—predominantly compressive and tensile—within complete dentures and affect the transmission of these stresses to the supporting soft and hard tissues [8,9,10,11,12].

Observing the extent and distribution of pressure on the contact surfaces of the acrylic teeth can help to understand the occurrence of damage to the complete dentures. To get as close as possible to the real situation, our work used maxillary and mandibular complete dentures obtained from the dental impression of a completely edentulous patient whose dental ridges were used to fabricate two identical pairs of dentures. Since the occlusal relationship plays an important role in the contact of the acrylic teeth, we will investigate the bite of complete dentures in eugnathic and progenic occlusal relationships both experimentally and numerically. The experimental investigations will give us an insight into the strength of the material of complete dentures, but also determine the magnitude of the compressive force at which damage occurs. Through numerical modelling and simulation, we will determine the magnitude and distribution of pressure on the contact surfaces of acrylic teeth at two forces of 100 N and 200 N, which can occur in edentulous individuals with complete dentures.

A full denture made of acrylic is an inexpensive service in dental practice, and It can be replaced relatively easily in the event of damage. To do this, it is necessary to know the causes that lead to damage over the course of many years.

## 2. Fabrication of Prostheses: 3D Scanning and 3D Models

An impression of the prosthesis was taken for a completely edentulous patient. Gnathostar factory acrylic teeth (Ivoclar Vivadent, Schan, Liechtenstein, Dentist & Dental Technician Blogs | Ivoclar, https://www.ivoclar.com/en_li/blog?page=1&limit=12&sorting=date%3Adesc&filters=%5B%5D, accessed on 19 May 2025) were used to insert the teeth into the full denture, and the heat-curing acrylic material ProBase HOT (Ivoclar Vivadent, Schan, Liechtenstein) was used to polymerise the base.

X-ray computed tomography (CT) was used to create a credible 3D model. To get as close as possible to the real case in the numerical simulations, the patient’s upper and lower complete dentures were scanned with the SORODEX Scanora 3D device (Biotech Inventions Ltd., Hong Kong, SCANORA 3D—Biotech Innovations Limited, http://www.biotech-in.com/soredex/scanora3d.html, accessed on 19 May 2025), as shown in Figure 2a for the upper prosthesis and in Figure 2b for the lower prosthesis. Each of the dentures was scanned several times in different positions so that the three-dimensional model of the upper and lower prosthesis could be determined as accurately as possible. After scanning, a rough 3D model was created, which, in addition to the prosthesis, also formed the framework, as shown in Figure 3a for the maxillary prosthesis and in Figure 3b for the mandibular prosthesis. In the modelling phase, the lower part of the framework was removed, and the prosthesis was remeasured in the 3D model.

## 3. Experimental Bite Testing of the Dentures

Prombonas and Dimitris [13] conducted experimental tests of maxillary and mandibular prostheses on hydraulic presses. They applied vertical occlusal forces in the range of 30 to 110 N and measured the principal stresses between the mandibular and maxillary prostheses in the range of −1.07 to 10.2 MPa.

Compression testing of the bite of complete dentures in the eugnathic and progenic relationship is used to determine the strength of the material of complete dentures to find a weak point, and also to determine the level of compressive force at which complete fracture of the denture occurs. The tests were carried out at a room temperature of +24.6 °C on an INSTRON 1255 servo–hydraulic press (Instron Ltd. High Wycombe, Buckinghamshire, UK) with a constant pressing speed v = 1 mm/min. The plaster moulds were made specially for complete dentures in eugnathic and, especially, in progenic occlusal relationships. Figure 4a shows the insertion of the dentures in the eugnathic relationship, while Figure 4b shows the insertion of complete dentures in the preogenic occlusal relationship before testing. Figure 5a shows the situation after the bite test of a complete denture in the eugnathic relationship, in which the acrylic base is first cracked between the first left central incisor and the first upper central acrylic incisor, and then longitudinally between the left cuspid and left acrylic first premolar. Figure 5b shows the situation after compression testing of a complete denture in a progenic relationship in which longitudinal cracks occurred on the acrylic teeth. During both tests, a force–displacement diagram was recorded, which is shown in Figure 6. Figure 6 shows that the force increased gradually, initially with a small slope, and then with a progressive increase in force. Due to the different occlusal relationships, there was a change in the slope after only 0.2 mm of displacement. The first change in the force–displacement curve, at which no damage was yet visible, occurred in the eugnathic relationship at a displacement of 0.83 mm and a force of 6.4 kN, and in the progenic relationship, at a displacement of 0.86 mm and a force of 7.01 kN. At a displacement of 1.39 mm and a force of 15.43 kN, the acrylic base between the first left central incisor and the first right central incisor cracked in the eugnathic relationship, and at a displacement of 1.41 mm and a force of 16.99 kN, the first longitudinal cracks appeared in the acrylic first right premolar and the second premolar. Figure 5a,b shows that the acrylic base did not detach from the acrylic tooth in any case. It can also be seen from the diagram in Figure 6 that at a force of 200 N, there was no drop in force in the two occlusal relationships, as the first changes on the smooth force curve only shifted at a force of 6400 N, i.e., at 32 times the force of the simulated bite. This confirms the hypothesis that the strength of the bond between the acrylic base and the acrylic tooth is not the weakest element in compression and has no influence on the load-bearing capacity of complete dentures in eugnathic or progenic occlusal relationships at a force of 200 N. The result of a good bond is that the acrylic base and the acrylic teeth break faster than the acrylic tooth, which detaches from the acrylic base. This finding allows us to consider only the material properties of the acrylic base and the acrylic teeth in the numerical modelling, without modelling their connection separately with finite elements.

Since experimental measurements do not allow for insight into the pressure distribution in the contact of acrylic teeth in complete dentures, we will use numerical simulations to monitor the stress and pressure distribution for the eugnathic and progenic occlusal relationships of complete dentures.

## 4. Numerical Modelling and Simulations

Numerical simulations according to FEM are used both in the process and in the determination of the strength and mechanical behaviour of the material [14,15,16]. Ateș et al. [17] used a two-dimensional finite element analysis to investigate the effect of the localisation of the occlusal contact on the stress distribution in complete maxillary denture bases. The results showed that the maximum compressive stresses are always concentrated at the transition to the artificial tooth base, regardless of the localisation of the occlusal contact. Manikani et al. [18] applied 3D finite element modelling and simulation of the residual ridge, mucosa, and prosthesis base in a coronal section created from the scanned and modelled prostheses of a patient. A vertical static load of 100 N was applied by the mandibular model to the maxillary model. The results showed a variation in the von Mises stress distribution between the canines and the one-piece teeth.

The scanned structure of the prosthesis is designed in great detail geometrically, as can be seen for the maxillary prosthesis in Figure 3a and the mandibular prosthesis in Figure 3b. A precise geometric–spatial CAD model was created using reverse engineering and computer-aided design. On the basis of a three-dimensional X-ray image (CT), the outer surfaces of the prosthesis and the teeth were produced with three-dimensional triangular tetrahedrons. Tetrahedral elements form the closed volumes of the prosthesis. The tetrahedrons were concentrated suitably at the points where we expect large changes in the stress and strain fields. With these elements, we can fill the inner volume of the prosthesis very precisely. Sets of tetrahedra describe the shape and mechanical behaviour of prostheses precisely in a numerical sense. Each tetrahedron is defined by the coordinates of four vertices, and these vertices form four triangles that delimit the volume of the tetrahedron. The numerical model consists of an upper and a lower prosthesis, as shown in Figure 7a for the upper prosthesis, with 651,236 elements and eight nodes (“CONTINUMUM 3D”), and in Figure 8 for the lower prosthesis, with 567,325 finite elements. For numerical accuracy, it is important that the tetrahedra are as regular as possible, which means that all sides should have approximately the same length. The prostheses are geometrically very diverse, so it was necessary to use many finite elements.

The acrylic teeth of the two dentures touch each other when biting down. The bite of complete dentures is simulated numerically in the eugnathic and progenic occlusal relationships. During the simulated bite, contact occurs between the teeth of the upper and lower dentures. Contact stresses and internal forces are generated at these points, which are transmitted via the teeth to the supporting structure of the denture and via the alveolar ridges to the jaw. The numerical spatial model of the maxillary and mandibular prostheses in a eugnathic relationship is shown in Figure 9, and the numerical spatial model of the maxillary and mandibular prostheses in a progenic relationship is shown in Figure 10.

The prostheses consist of an acrylic base and teeth with different mechanical properties. The mechanical properties were measured experimentally and described in Pavlin et al. [19], while the analysis of bond strength was described in Pavlin et al. [8].

The results of the experimental measurements for the PMMA prosthesis acrylic base and a factory-made acrylic prosthesis of a modulus of elasticity and a Poisson’s ratio are given in Table 1. Their average values were used for the FEM simulations [19]. The mechanical properties that we measured experimentally ourselves [19] were in the range of the values used in the numerical simulations of Gonzalez-Martin et al. [4]. The results of the analysis of the bond strength between the acrylic teeth and the acrylic base of a complete denture show that we can strengthen the bond in a thin zone between the acrylic and the acrylic teeth before machining the underside of the acrylic teeth. With the aim of determining the pressure distribution between the acrylic teeth, a numerical simulation was carried out at a compressive force of 100 N and 200 N of complete dentures in eugnathic and progenic occlusal relationships. The calculation was performed with the Simulia Abaqus 13.0 program package (Dessault Systems, Vélizy-Villacoublay, France) [20].

The challenge in the calculation of numerical simulations is the contact of uneven surfaces, where the local pressure can reach values that exceed the strength of the material, while the neighbouring tetrahedral element can be completely pressureless. Therefore, the equivalent von Mises stresses are calculated for the stress distribution in the material, which indicate the stresses in the elements, both in the contacting and in the neighbouring elements, and, thus, in the entire volume of the prosthesis. Figure 11 shows the distribution of the equivalent von Mises stresses in the eugnathic relationship while Figure 12 shows the same in the progenic relationship over the surface of the model at bite forces of 100 N and 200 N. The comparison of the figures shows that in the eugnathic relationship, the tips of the acrylic teeth in the upper denture are less stressed, while in the progenic relationship, the stresses at the tips of the acrylic teeth are high. Figure 11 and Figure 12 show the same scale for the stresses. Figure 11 shows that the stresses in the prosthesis in the eugnathic bite position are in the area between the acrylic base and the angled acrylic teeth, and in the extended bite position, the maximum stresses are in the area of the acrylic 3 upper (L+D) and 2 and 3 lower left teeth. The described locations of the maximum stresses correspond with the experimental tests, as shown in Figure 4 and Figure 5. In the bite test of complete dentures in the eugnathic position, the acrylic base between two acrylic teeth cracked first, whereas in the bite test of complete dentures in the prognathic position, the dental inserts cracked longitudinally. The scales in all the figures are the same, with the highest stresses being up to 20 MPa. The comparison between the image at a bite force of 100 N and 200 N shows that the stresses increase and the area on the acrylic base increases due to the contact of the acrylic teeth.

Figure 13 shows the distribution of the equivalent von Mises stresses on the surface of the prosthesis and the acrylic teeth in the contact area in the eugnathic occlusion relationship at a bite force of 100 N and 200 N for the upper and lower prostheses. Figure 13a shows the maxillary prosthesis, Figure 13c shows the mandibular prosthesis at a bite force of 100 N, Figure 13b shows the maxillary prosthesis, and Figure 13d shows the mandibular prosthesis at a bite force of 200 N.

Figure 13 shows a representation of the distribution of equivalent stresses in the contact area in the eugnathic occlusal relationship at a bite force of 100 N and 200 N for maxillary and mandibular dentures.

Figure 13 shows the redistribution of the maximum stresses on the contacting tooth surfaces in such a way that the pressure at the original contact points increases gradually, while at the new contact surfaces, the values increase more rapidly and the contact area increases more quickly. It can be said that the architecture of the complete denture is adapted flexibly when the bite force is increased so that the contact surfaces increase, which reduces the increase in pressure between the acrylic teeth.

Figure 14 shows the von Mises equivalent stresses on the surface of the contacting teeth in the progenic occlusal relationship at bite forces of 100 N and 200 N for the upper and lower dentures. Figure 14a shows the maxillary prosthesis, Figure 14b shows the mandibular prosthesis at a bite force of 100 N, Figure 14c shows the maxillary prosthesis, and Figure 14d shows the mandibular prosthesis at a bite force of 200 N.

Figure 14 shows that in a progenic occlusal relationship, there is practically no redistribution of stresses across the contact surfaces, but rather that the stress values and the contact surface increase.

With a full denture in eugnathic contact, the redistribution occurs in such a way that the acrylic base absorbs the stresses via the acrylic teeth, and, in the course of adaptation, more and more lateral acrylic teeth take over the contact pressures, while the pressure increase in the incisors decreases.

With a full denture in progenic contact, the redistribution takes place in such a way that the acrylic base also absorbs the stresses via the acrylic teeth, but when the stress is transferred to the lateral acrylic teeth, the canines and the tips of the incisors and molars remain under the greatest stress.

Figure 14 presents an illustration of the distribution of the equivalent stresses in the contact area in the progeny–occlusal relationship at bite forces of 100 N and 200 N for maxillary and mandibular dentures. The equivalent von Mises stresses show the stresses in the material structure and between the acrylic base and the acrylic teeth. They, therefore, indicate the points on the denture where a crack in the denture is to be expected, or where damage has occurred. In order to understand the contact relationship between the acrylic teeth and thus the cracking of the acrylic teeth in a progressive relationship, the pressure between the upper and lower dentures must be considered.

Figure 15 shows the pressure distribution of a complete denture in eugnathic patients, and Figure 16 shows the progenic occlusal relationship. Figure 15a shows the pressure distribution on the surface of the upper prosthesis at a bite force of 100 N, and Figure 15b shows the same at a bite force of 200 N. The highest pressure distribution can be seen in Figure 15c at 100 N and in Figure 15d at a bite force of 200 N on the lower prosthesis. Figure 15c,d shows that the highest pressures occur near the contact with the acrylic teeth of the lower denture, and that the pressures are distributed over the surface of the dental contacts when the acrylic base of the denture deforms.

Figure 16a shows the pressure distribution over the surface of the upper prosthesis with a bite force of 100 N, and Figure 16b shows the same with a bite force of 200 N. The highest pressure distribution can be seen in Figure 16c at 100 N and in Figure 16d at a bite force of 200 N on the lower prosthesis. Figure 16c,d shows that the highest pressures occur in the area of contact with the acrylic teeth. In contrast to the pressure distribution over the surface with deformation of the acrylic base in the eugnathic relationship, in full dentures in the progenic relationship, the pressures are redistributed over the surface of the acrylic teeth in such a way that the pressure and the contact area on the tooth increase without the acrylic base being deformed or adjusted significantly. This result confirms the experimental behaviour of the prosthesis in the progenic relationship, where a higher maximum force was achieved before the acrylic teeth fractured.

## 5. Discussion

The limit load capacity of complete dentures in eugnathic and progenic occlusal relationships was tested experimentally. We tested the same dentures that we used with alabaster moulds in the eugnathic and progenic occlusal relationships. The test results showed that for full dentures in the eugnathic relationship, the acrylic base between the first left and first right and the acrylic third left and fourth left cracked longitudinally.

The experimentally determined load curve shows the fit of the acrylic teeth with an increasing force of the two complete dentures until the first slip. The first slip leads to a local drop in force, with the first damage occurring in the area of the same compressive force values of 6.14 kN for the eugnathic relationship and 7.01 kN compressive force for the progenic relationship. Although there were different fracture types of complete dentures in both eugnathic and progenic occlusal relations, the maximum bite force values did not differ significantly. At the maximum compressive forces, the acrylic base fractured in the eugnathic relationship, and the acrylic teeth fractured longitudinally in the progenic occlusal relationship. The cracking of the acrylic teeth or the acrylic base depends on whether the complete dentures are in a eugnathic or prognathic relationship, but in no case did the acrylic teeth detach from the acrylic base. This confirms the hypothesis that the adhesive force of the acrylic base to the acrylic teeth is not a critical point for damage and that special numerical modelling of the adhesive force is not required. In this present study, a numerical simulation of the complete dentures in both eugnathic and progenic occlusal relationships was performed with two bite forces of 100 N and 200 N, which were taken as the reference values for edentulous patients in accordance with the literature [4,13,18]. The geometry of the complete dentures determined with 3D-computed tomography was taken from an edentulous patient whose edentulous ridges were used to fabricate two identical pairs of dentures. The results of the numerical simulations in both occlusal relationships showed that there is a redistribution of pressure on the contact surfaces during occlusion.

The results of the numerical simulations and other authors, such as Marti-Vigil et al. [21] and Kupprano et al. [22], showed that the von Mises stresses on the surface reach local values of the same rank with similar mechanical behaviour. Prombonas and Dimitris [23] conducted experimental tests of maxillary and mandibular prostheses on hydraulic presses. They applied vertical occlusal forces in the range of 30 to 110 N and measured the principal stresses between mandibular and maxillary prostheses in the range of −1.07 to 10.2 MPa. Grachev et al. [24] used FEM for the computer simulation and prediction of the longevity of removable complete dentures. It was found that both the displacement and the inclination of the individual tooth blocks have a negative influence on the cyclic durability of the dentures.

Ravi et al. [25] also applied vertical loads of up to 100 N to test prostheses and measured the applied strains. It was found that a tensile load prevailed in the maxillary denture while a compressive load prevailed in the posterior palatal region, but the external placement of the maxillary teeth caused a significant decrease in the compressive load.

With a full denture in eugnathic contact, redistribution occurs in such a way that the acrylic base takes over the stresses from the acrylic teeth, and, during adaptation, more and more angular dental inserts take over the contact pressures, while the increase in pressure on the incisors decreases. The von Mises equivalent stresses show that the maximum stresses are in the material structure and between the acrylic base and the acrylic teeth. This allows us to find points on the prosthesis where we can expect a crack in the prosthesis or points of damage in the eugnathic occlusal relationship.

With a complete denture in progenic contact, the redistribution is such that the load is transferred to the lateral acrylic teeth, and thus the canines and the tips of the incisors and molars remain the most heavily loaded.

The numerical results presented provide insight into the state of pressure redistribution between the dental contact surfaces and the transfer of load to the acrylic base, thus explaining the relationships that can lead to cracks in the acrylic base or damage to the acrylic teeth.

The fraction of the volume with von Mises stress above 20 MPa in relation to eugnathic and progenic occlusal relationships is shown in Table 2. It is evident that the fraction of the volume with stress equal to or greater than 20 MPa is approximately two times greater in progenic relationships than in eugnathic relationships.

Although the test results showed no significant difference in the static failure strength of total dentures in eugnathic and progenic conditions, the stress behaviour of dentures in eugnathic and progenic conditions is different.

## 6. Conclusions

In both eugnathic and progenic occlusal relationships, the overall load capacity of complete dentures was similar, but the pattern of stress distribution and damage differed. In the eugnathic occlusion, the highest stresses were observed in the acrylic base, increasing the risk of fatigue cracks and structural failure in that region. In the progenic occlusion, stress was concentrated on the acrylic teeth, particularly the canines and incisors, leading to longitudinal tooth fractures.

The numerical simulations confirmed that pressure redistribution plays a key role in the location and type of damage, depending on the occlusal contact pattern. To reduce the risk of denture failure, reinforcement of the acrylic base is recommended for eugnathic cases, while occlusal adjustments may help protect the teeth in progenic relationships.

Clinical examinations, as well as the experimental results, showed that in the case of a eugnathic bite, cracking occurs between the upper dental inserts 1 and 2 on the left and right, and very rarely between the upper units. The benefit of these results is that it is possible to access the local strengthening of only the inserts in the place of the dental incisors, and not the entire prosthesis, as well as from the inside in an acrylic base, which does not affect the aesthetics and increases the working life of the prosthesis.

## Figures and Tables

**Figure 1 materials-18-02427-f001:**
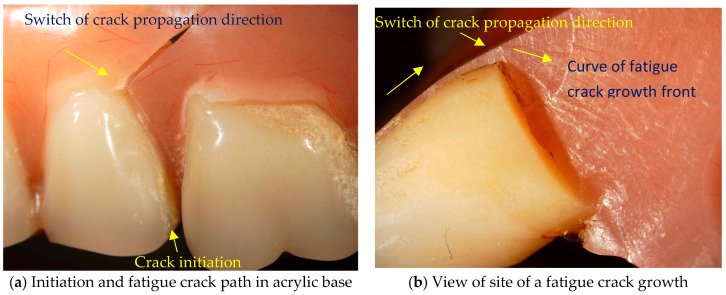
Location of a fatigue crack in the area between the cuspid and first premolar acrylic teeth.

**Figure 2 materials-18-02427-f002:**
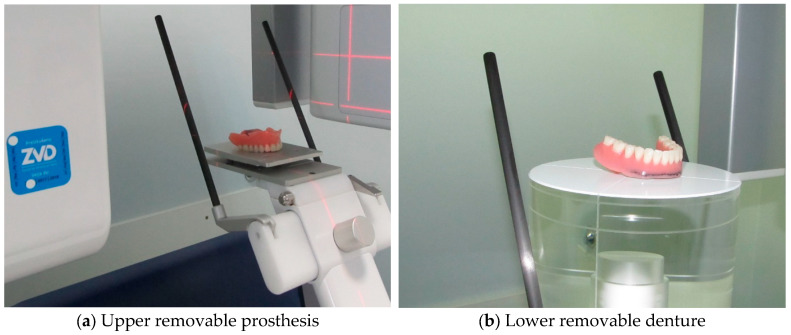
Imaging of the upper and lower complete dentures.

**Figure 3 materials-18-02427-f003:**
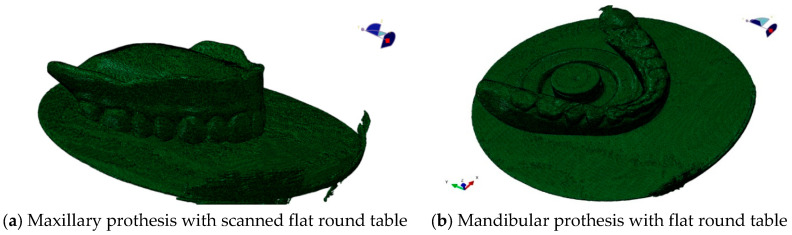
Spatial model of the scanned upper and lower dentures.

**Figure 4 materials-18-02427-f004:**
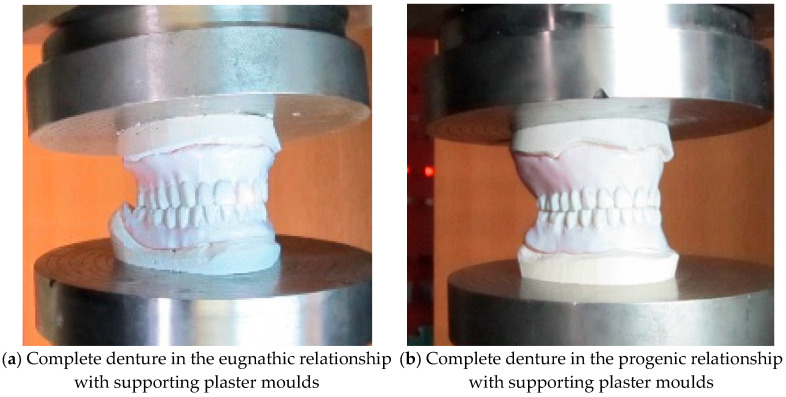
Experimental setup for denture compressive testing.

**Figure 5 materials-18-02427-f005:**
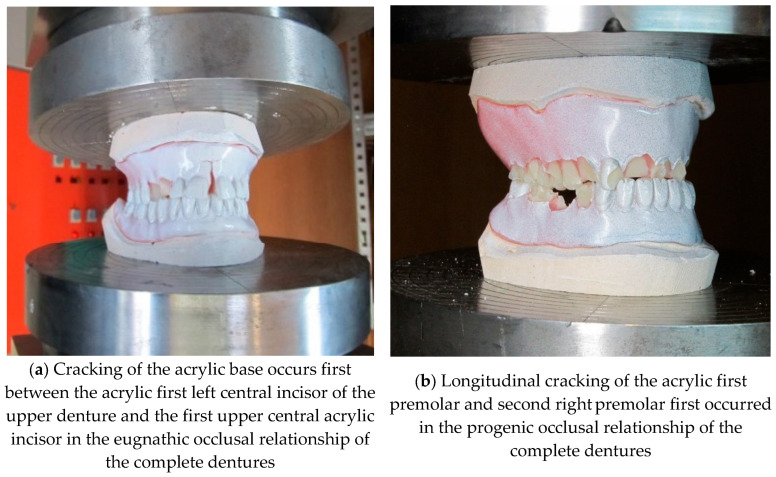
Dentures’ appearance after compressive testing.

**Figure 6 materials-18-02427-f006:**
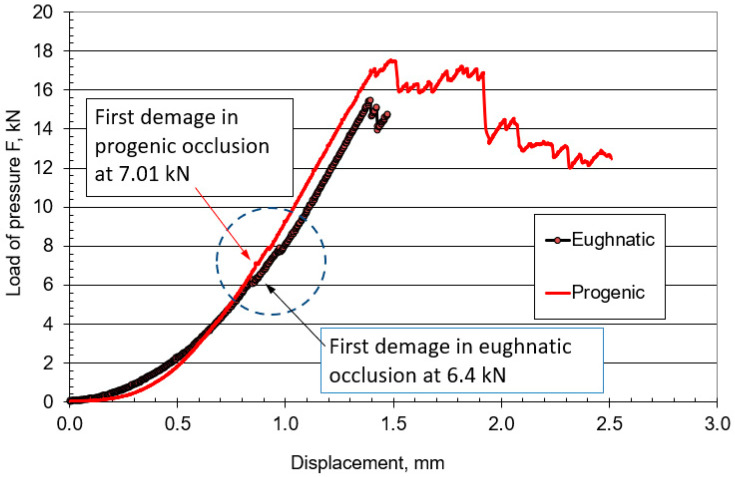
Recorded force–displacement diagram during the bite testing of complete dentures in eugnathic and progenic occlusal relationships.

**Figure 7 materials-18-02427-f007:**
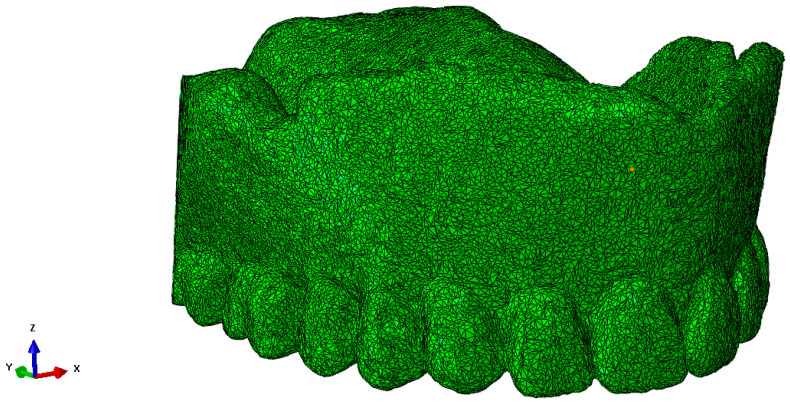
The upper part of the denture with 651,236 finite elements after scanning.

**Figure 8 materials-18-02427-f008:**
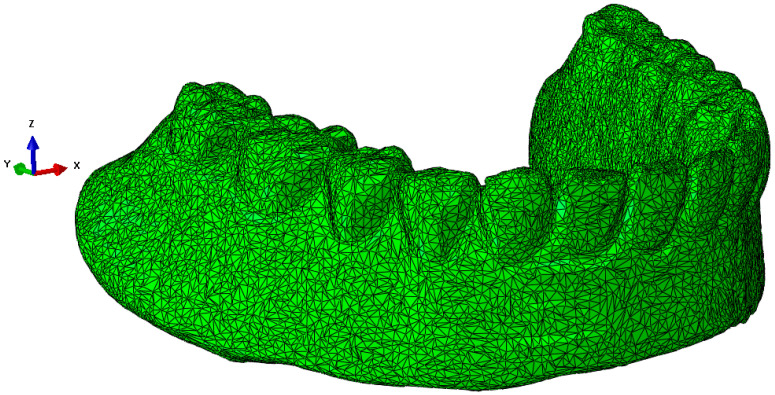
The lower part of the denture with 567,325 finite elements.

**Figure 9 materials-18-02427-f009:**
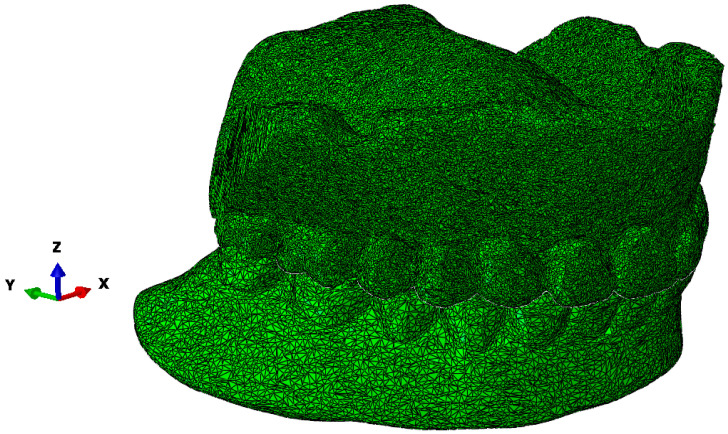
Numerical model of upper and lower dentures in the eugnathic occlusal relationship.

**Figure 10 materials-18-02427-f010:**
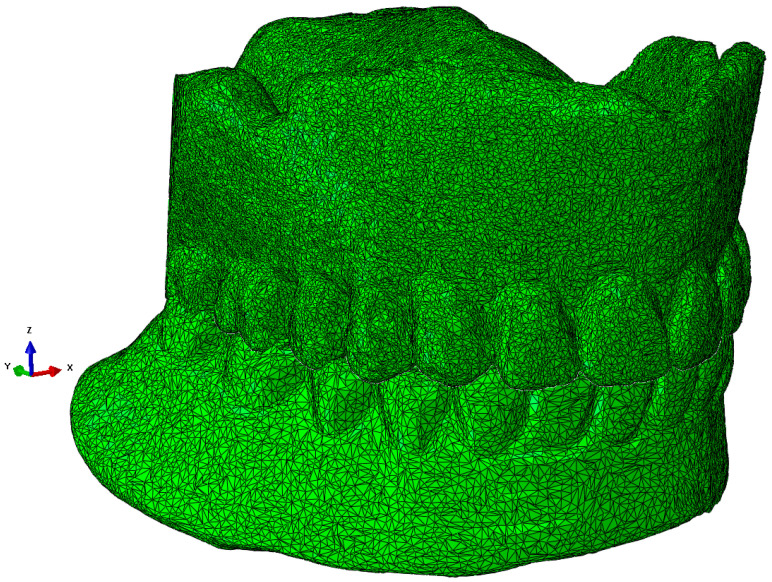
Numerical model of the upper and lower dentures in the progenic occlusal relationship.

**Figure 11 materials-18-02427-f011:**
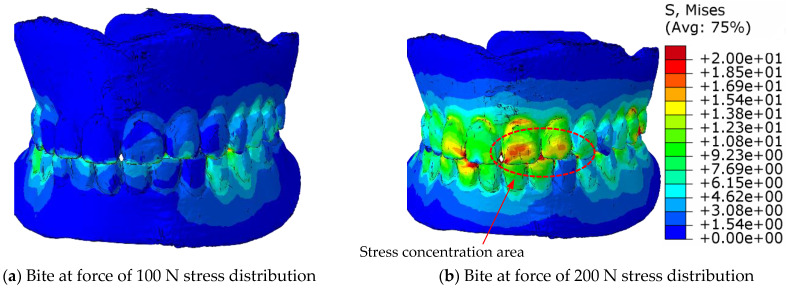
Stress display of the equivalent stresses of the von Mises distribution on the surface of the denture in the eugnathic occlusal relationship.

**Figure 12 materials-18-02427-f012:**
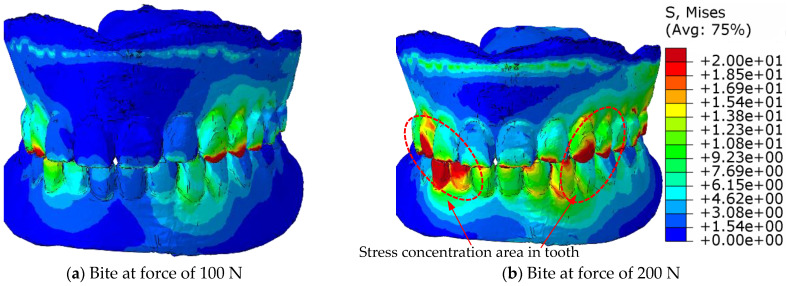
Presentation of stress equivalent stresses on the surface of the dentures in the progenic occlusal relationship for stress from zero to 20 MPa.

**Figure 13 materials-18-02427-f013:**
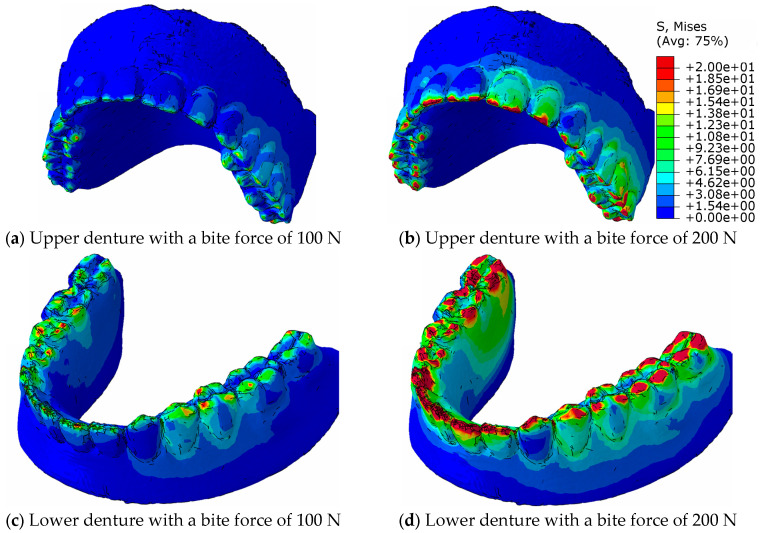
Presentation of the distribution of equivalent stresses in the contact area in the eugnathic occlusal relationship with bite forces of 100 N and 200 N for upper and lower dentures for stress from zero to 20 MPa.

**Figure 14 materials-18-02427-f014:**
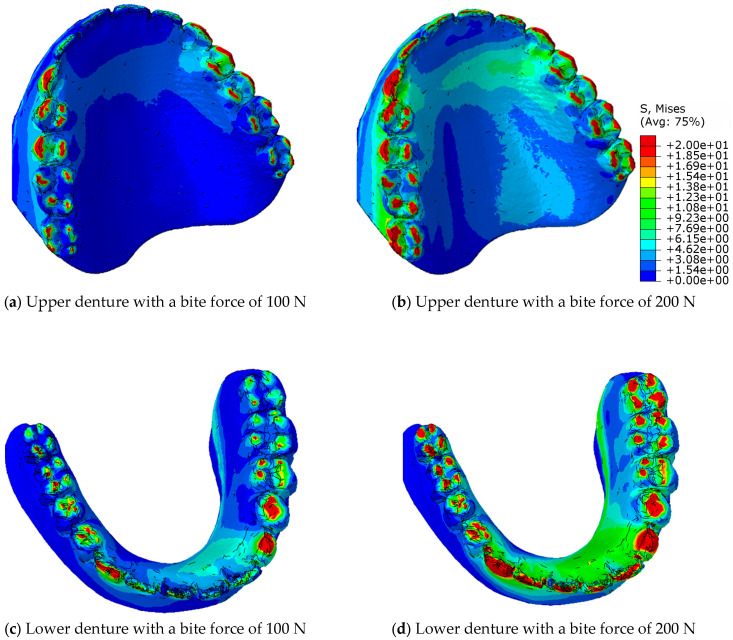
Presentation of the distribution of equivalent stresses in the area of contact in the progenic occlusal relationship with a bite force 100 N and 200 N for upper and lower dentures for stress from zero to 20 MPa.

**Figure 15 materials-18-02427-f015:**
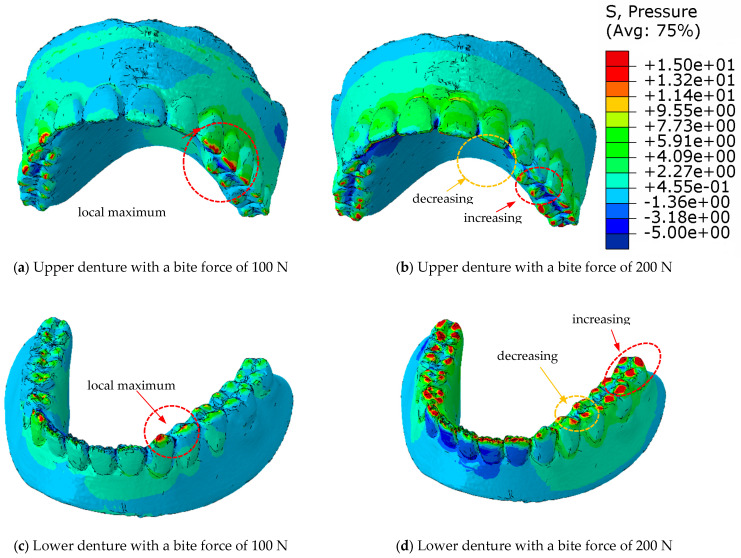
Pressure distribution of complete dentures in a eugnathic occlusal relationship.

**Figure 16 materials-18-02427-f016:**
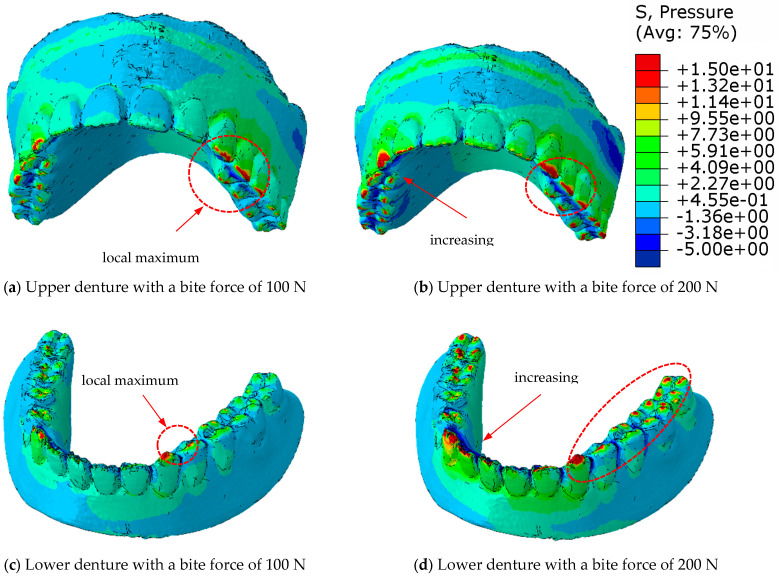
Pressure distribution of complete dentures in the progenic occlusal relationship.

**Table 1 materials-18-02427-t001:** Mechanical properties of the acrylic materials used in FEM simulations [19].

	Modulus of Elasticity E, MPa	Poisson Number ν, -
PMMA acrylic base	4151	0.33
PMMA in factory-made acrylic tooth	6315	0.30

**Table 2 materials-18-02427-t002:** The fraction of the volume with von Mises stress above 20 MPa in relation to eugnathic and progenic occlusal relationships.

	Bite Force	Von Mises Stress, ≥20 MPa
Eugnathic	100 N	0.0071%
	200N	0.0352%
Progenic	100N	0.0148%
	200N	0.0710%

## Data Availability

The data presented in this study are available on request from the corresponding author.

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
