# Peer review of "Experimental Testing and Numerical Bite Simulation of Complete Acrylic Dentures in Eugnathic and Progenic Occlusal Relationships"

_materials, 2025, doi:10.3390/ma18112427_

Round 1
Reviewer 1 Report
Comments and Suggestions for Authors
The manuscript titled “Experimental Testing and Numerical Bite Simulation of Complete Acrylic Dentures in Eugnathic and Progenic Occlusal Relationships” addresses a biomechanically and clinically relevant topic, especially important in the context of prosthetic planning for edentulous patients with varying skeletal and occlusal anatomies. The study presents a dual methodology—experimental load testing and numerical simulation using the finite element method—to evaluate stress distribution and fracture behavior in acrylic dentures under different occlusal schemes. From the perspective of a practicing oral surgeon, this work is commendable for integrating real patient anatomy into numerical simulation. The use of 3D tomography to reconstruct denture geometry from actual impressions (as outlined between lines 100 and 104) offers realism and anatomical accuracy that many simulation-based studies lack. This approach provides a model that is not only theoretically sound but also clinically translatable, especially when dealing with post-extraction prosthetic rehabilitation in patients with altered occlusal parameters such as Class III skeletal relationships. The results of experimental compressive testing indicate that structural failure of complete acrylic dentures occurs at high loading thresholds—6400 N for eugnathic and 7010 N for progenic occlusion. These thresholds, significantly higher than physiologic masticatory forces in edentulous patients (100–200 N), reinforce the mechanical resilience of current denture designs under static load. Nevertheless, the findings also reveal that fracture initiation patterns differ based on occlusal type—base fractures dominate in eugnathic relationships while longitudinal fractures in the acrylic teeth are more common in progenic cases. This distinction has immediate clinical implications for material reinforcement, occlusal equilibration, and long-term wear resistance of prostheses, especially in patients with skeletal discrepancies. The numerical simulations reinforce the experimental findings. von Mises stress distributions reveal that in eugnathic occlusion, stress tends to redistribute efficiently across lateral teeth and the acrylic base. In contrast, the progenic setup shows concentrated stress at the tips of incisors and canines, resulting in less effective pressure dispersion. Clinically, this suggests a greater risk of tooth fracture and potential fatigue damage in progenic occlusion—a finding that supports reinforcement or design modifications in such cases.
Language throughout the manuscript, while mostly comprehensible, requires moderate revision for clarity and scientific tone. Several instances of long, unstructured sentences and ambiguous phrasing reduce the manuscript’s readability. For example, in lines 23–25, the sentence “numerical simulations… carried out with bite forces of 100 N and 200 N, whereby no damage occurred during the experiment” should be rephrased for precision and flow. Terminological consistency is another concern—terms like “progenital” appear sporadically and inaccurately (line 330 onward) and should be corrected to “progenic” or “prognathic” throughout the manuscript. The clinical interpretation of results could be improved by addressing their practical applications more explicitly. The discussion would benefit from guidance on whether certain occlusal relationships demand routine reinforcement, special occlusal schemes, or relining protocols. Additionally, the manuscript currently misses the opportunity to connect the biomechanical findings to daily prosthodontic planning—especially in preprosthetic surgical preparation, where the surgeon’s understanding of force distribution can guide ridge preservation or recontouring. Figures 11 through 16 provide strong visual support, but captions should be more descriptive for non-specialist readers. Statements such as “a) at force 100 N” are too brief. Expanding these to include context (e.g., denture type, anatomical location, and implication of results) would increase their value. The manuscript would also benefit from a condensed schematic comparing both occlusal relationships in terms of maximum stress zones and potential fracture risks. Overall, this is a robust and well-executed study with tangible clinical relevance. As a clinician involved in both surgical and prosthetic aspects of oral rehabilitation, I see strong merit in the conclusions, particularly in how they support the need for occlusal individualization and material resilience in denture fabrication. With moderate language correction and greater emphasis on clinical applicability, the manuscript should be suitable for publication and will serve as a valuable resource for both researchers and practitioners involved in complete denture therapy.
Comments on the Quality of English LanguageThe manuscript is generally understandable and the technical terminology is used appropriately. However, there are several instances where sentence structure and phrasing could be improved for clarity and fluency. Some sentences are overly long or contain awkward constructions, and minor grammatical issues occasionally interrupt the flow of the text. Additionally, the use of the term “progenic” is not standard in the dental literature and may confuse readers unless clearly defined. A thorough language review by a native English speaker or professional editor is recommended to ensure that the scientific content is communicated more clearly and precisely.
Author Response
Thank you for your comments and suggestions. We have considered and provide answers to your questions as follows:
- “Language throughout the manuscript, while mostly comprehensible, requires moderate revision for clarity and scientific tone. Several instances of long, unstructured sentences and ambiguous phrasing reduce the manuscript’s readability. For example, in lines 23–25, the sentence “numerical simulations… carried out with bite forces of 100 N and 200 N, whereby no damage occurred during the experiment” should be rephrased for precision and flow.”
Answer: We corrected the above sentence and had the entire paper reviewed by a native English speaker.
- Terminological consistency is another concern—terms like “progenital” appear sporadically and inaccurately (line 330 onward) and should be corrected to “progenic” or “prognathic” throughout the manuscript.
Answer: Throughout the paper, “progenital” and “prognathic” are replaced with the correct term “progenic”.
- The clinical interpretation of results could be improved by addressing their practical applications more explicitly.
Answer: According to this comments we have put in introduction:
Patients come for an examination twice a year. After two years of using complete dentures, we find that the patient's used dentures show damage on the surface of the upper and lower acrylic teeth. Less often than damage, cracks or breakage of the acrylic base of the upper prosthesis appear. The lower denture cracks less often. Patients report that when a crack occurs, its spread is relatively fast and within two to three days. There can be a complete dysfunction or fracture of the upper denture. Damage to the occlusal surface of acrylic teeth occurs more often in patients with improper progenic bite, when biting hard food and due to lifestyle habits.
- The discussion would benefit from guidance on whether certain occlusal relationships demand routine reinforcement, special occlusal schemes, or relining protocols.
Answer: In the paper we have redistributed discussion and conclusion. Now discussion and conclusion are more precise. We have written in discussion follow:
The results of numerical simulations and other authors Marti-Vigil et al [21] and Kupprano et al [22] showed that the von Mises stresses on the surface reach local values of the same rank with similar mechanical behaviour. Prombonas and Dimitris [23] conducted experimental tests of maxillary and mandibular prostheses on hydraulic presses. They applied vertical occlusal forces in the range of 30 to 110 N and measured principal stresses between mandibular and maxillary prostheses in the range of -1.07 to 10.2 MPa. Grachev et al [24] used FEM for computer simulation and prediction of the longevity of removable complete dentures. It was found that both the displacement and the inclination of the individual tooth blocks have a negative influence on the cyclic durability of the dentures.
Ravi et al [25] also applied vertical loads of up to 100 N to test prostheses and measured the applied strains. It was found that a tensile load prevailed in the maxillary denture, while a compressive load prevailed in the posterior palatal region, but the external placement of the maxillary teeth caused a significant decrease in the compressive load.
- Additionally, the manuscript currently misses the opportunity to connect the biomechanical findings to daily prosthodontic planning—especially in preprosthetic surgical preparation, where the surgeon’s understanding of force distribution can guide ridge preservation or recontouring.
Answer: We have made reconstruction of paper and extened chapter Discussion and make shorter Conclusion.
- Figures 11 through 16 provide strong visual support, but captions should be more descriptive for non-specialist readers. Statements such as “a) at force 100 N” are too brief.
Answer:
We have added a more detailed description of how the stresses s are generated and supplemented it with a unified legend and description of the individual images from Fig. 11 to Fig. 16.
In the FEM simulation, the upper surface of the maxillary prosthesis and the lower surface of the mandibular prosthesis are pressed against the alabaster mold with force. First with 100 N and then with 200 N. When pressed, the force changes into pressure and tension occurs in the model.
We have also added a table that shows the percentage increase in tension in the prosthesis at each bite force.
The fraction of the volume with von MIses stress above 20 MPa in relation to eugnathic and progenic occlusal relationships is shown in Table 2. It is evident that the fraction of the volume with stress equal to or greater than 20 MPa is approximately two times greater in progenic relationships than in eugnathic relationships.
Table 2: The fraction of the volume with von MIses stress above 20 MPa in relation to eugnathic and progenic occlusal relationships
|
Bite force |
Von Mises stress, >20 MPa |
Eugnathic |
100 N |
0.0071% |
|
200N |
0.0352% |
Progenic |
100N |
0.0148% |
|
200N |
0.0710% |
- Expanding these to include context (e.g., denture type, anatomical location, and implication of results) would increase their value.
Answer: We have put in conclusion:
Clinical examinations as well as experimental results show that in case of eugnathic bite, cracking occurs between the upper dental inserts 1 and 2 on the left and right, and very rarely between the upper units. The benefit of these results is that it is possible to access the local strengthening of only the inserts in the place of the dental incisors, and not the entire prosthesis. And that from the inside in an acrylic base, which does not affect the aesthetics, and increases the working life of the prosthesis.
- The manuscript would also benefit from a condensed schematic comparing both occlusal relationships in terms of maximum stress zones and potential fracture risks.
Answer: We have said that in the Eugnathic relationship, von Mises stresses are important, and in the Progenic relationship, pressures are important, which are also distributed according to the construction of the dentures in the bite.
- Comments on the Quality of English Language
The manuscript is generally understandable and the technical terminology is used appropriately. However, there are several instances where sentence structure and phrasing could be improved for clarity and fluency. Some sentences are overly long or contain awkward constructions, and minor grammatical issues occasionally interrupt the flow of the text. Additionally, the use of the term “progenic” is not standard in the dental literature and may confuse readers unless clearly defined. A thorough language review by a native English speaker or professional editor is recommended to ensure that the scientific content is communicated more clearly and precisely.
Answer: We have asked native English speaker for final English correction.
In addition to correcting the article for native English speakers, we have added in the introductory part of the article:
A proper bite is a eugnathic relationship. In this relationship, the upper teeth, the incisors, cover the lower incisors and slide along the surface of the front lower teeth until the premolars and molars come into contact with each other. The molars are the teeth for chewing food.
A progenic relationship is the contact of the lower and upper incisors, and occurs when the lower jaw moves forward, so that there is no overlap, but rather there is first contact of the tips of the upper and lower teeth. Because of the contact of the front incisors, the molars do not fit completely and

Reviewer 2 Report
Comments and Suggestions for Authors
This manuscript addresses an important and relevant topic in dental biomechanics. The combined use of computed tomography-based modeling, finite element simulations, and mechanical testing enhances the robustness of the study. However, the manuscript requires improvements in organization, clarity of language, and consistency in terminology. Below are detailed comments.
The abstract contains long sentences and few logical connectors, which hinders reading fluency. I recommend revising the text to make it more concise and direct. To achieve this, reorganize the abstract into the following structure: brief introduction to the problem, objective, methods, results, and conclusion addressing the study’s aim.
In the introduction, the first paragraph is confusing; please review the main idea to be addressed. At this point, prioritize the contextualization of the topic to highlight the problem investigated in the study. The opening sentence of the introduction, “Dentures are an important part of the diet of patients who have lost their own teeth,” is unclear, and its actual meaning is difficult to understand. Please rewrite it.
The study's objective is presented in both the second and third paragraphs of the introduction. These should be combined and rewritten to ensure conciseness and avoid redundancy. I suggest including the objective in the final paragraph of the introduction.
The references are insufficiently explored in the introduction. They appear concentrated in simple sentences and need more thorough discussion—for example: “Although there are studies [8–12] on the strength of the bond between acrylic teeth and acrylic base,...” I recommend elaborating on the results and conclusions of the cited studies.
Be clear in the introduction to enable the reader to understand the study’s applicability, originality, relevance, and innovation.
In the Methodology, regarding the sentence “An impression of the prosthesis was taken for a completely edentulous patient,” it suggests that a patient was involved. Therefore, it would be necessary to indicate approval from an ethics committee and the signing of an informed consent form. No reference to this important aspect is made in the text.
The workflow is well-structured, but some descriptions (mesh properties, boundary conditions, load application in simulations) require greater detail and standardization to ensure reproducibility.
Specify material properties clearly in a table, with sources or testing methods.
Figures are informative, but some captions lack sufficient explanation of what is being shown. For instance, in Figures 11–16, include details on scale bars, color meaning, and model orientation.
Some figures seem repeated or disorganized, compare the presentation of figure 14 with the rest of the results. Standardize the figures.
About the results and discussion sections, the comparison between eugnathic and progenic relationships is well developed. However, numerical data should be more systematically presented, possibly in tables.
The discussion could be enriched by referring to the clinical implications in more depth. How these findings affect prosthodontic planning or material selection?
Conclusions are comprehensive but could be more succinct. I suggest to emphasize the clinical and biomechanical implications rather than repeating detailed findings.
Author Response
Thank you for your comments and suggestions. We have considered and provide answers to your questions and we think improved our paper. We have correted paper and provided our answers as follows:
- “The abstract contains long sentences and few logical connectors, which hinders reading fluency. I recommend revising the text to make it more concise and direct. To achieve this, reorganize the abstract into the following structure: brief introduction to the problem, objective, methods, results, and conclusion addressing the study’s aim. “
We have rewrite new abstract as follows:
Abstract
Complete dentures are exposed to complex masticatory forces that may lead to material fatigue and eventual structural failure. Occlusal relationships, such as eugnathic and progenic, significantly influence the distribution of these forces. Understanding their biomechanical impact is essential for improving denture design and longevity.
The aim of this study was to evaluate the mechanical behaviour of complete dentures under bite loads in eugnathic and progenic occlusal relationships, using both experimental testing and numerical simulations. The focus was placed on identifying the conditions that lead to initial damage and the patterns of stress distribution.
The material properties of the denture base and artificial teeth were determined through experimental tensile and compressive testing on cylindrical PMMA specimens. Denture geometry was acquired via 3D tomography based on impressions of an edentulous patient. Experimental testing of the denture bite was conducted to determine the force thresholds at which initial cracks occur. Numerical simulations were carried out using finite element analysis at bite loads of 100 N and 200 N in both occlusal types, incorporating the obtained material parameters.
Experimental results showed that the first signs of denture damage occurred at 6400 N in eugnathic occlusion and 7010 N in progenic occlusion. Numerical simulations confirmed that pressure during occlusion is redistributed across multiple contact points, with broader distribution reducing localized stress. This redistribution was more efficient in eugnathic occlusion, which reduced the risk of longitudinal cracking in acrylic teeth. In contrast, progenic occlusion showed higher susceptibility to fractures within the acrylic denture base, particularly between adjacent teeth.
Both experimental and numerical approaches demonstrated that occlusal relationships directly affect the mechanical resilience of complete dentures. The findings highlight that eugnathic occlusion offers biomechanical advantages in stress distribution, potentially reducing the risk of fracture. Incorporating occlusal analysis into denture design protocols can enhance clinical outcomes and improve prosthetic longevity.
- In the introduction, the first paragraph is confusing; please review the main idea to be addressed. At this point, prioritize the contextualization of the topic to highlight the problem investigated in the study. The opening sentence of the introduction, “Dentures are an important part of the diet of patients who have lost their own teeth,”is unclear, and its actual meaning is difficult to understand. Please rewrite it.
Answer: We deleted this sentence and replaced it with:
Partial or complete edentulism negatively affects masticatory efficiency, speech articulation, and facial aesthetics; the fabrication of partial or complete removable dentures serves as a rehabilitation intervention that restores oral function, phonetic clarity, and a natural facial profile, significantly improving the patient's quality of life.
- “The study's objective is presented in both the second and third paragraphs of the introduction. These should be combined and rewritten to ensure conciseness and avoid redundancy. I suggest including the objective in the final paragraph of the introduction.”
Answer: The aim of study we have put at the end of rewritten introduction:
Although a full denture made of acrylic is an inexpensive service in dental practice, it can be replaced relatively easily in the event of damage. To do this, it is necessary to know the causes that lead to damage over the course of many years.
- “The references are insufficiently explored in the introduction. They appear concentrated in simple sentences and need more thorough discussion—for example: “Although there are studies [8–12] on the strength of the bond between acrylic teeth and acrylic base,...”I recommend elaborating on the results and conclusions of the cited studies. Be clear in the introduction to enable the reader to understand the study’s applicability, originality, relevance, and innovation.”
Answer: In accordance with your suggestion, the references that examine the strength of the bond between acrylic teeth and acrylic base are not related to the goals and results of this study, where experimental testing and numerical simulations investigate the load behavior on complete dentures in eugnathous and progenous occlusal relationships. Therefore, we removed references from 8-12 and wrote a text paragraph on the topic of bite force transfer to complete dentures, and we listed new references from 8-12 with an adequate topic.
We have put follows:
Acrylic teeth and the denture base, as the primary load-bearing components of complete dentures, are subjected to various functional stresses, including compression, tension, shear, and torsion, which may lead to fracture. Additionally, mechanical factors such as variable magnitudes and directions of masticatory loading, differences in the viscoelastic properties and thickness of the oral mucosa, absence of a precise axis of symmetry, non-uniform denture base thickness, and anatomical variations in the edentulous alveolar ridges of the maxilla and mandible complicate the clinical study of this topic. These factors also influence the distribution and behavior of internal stresses—predominantly compressive and tensile—within complete dentures and affect the transmission of these stresses to the supporting soft and hard tissues (8–12).
- Prombonas A, Poulis NA, Yannikakis SA. The Impact of Notches on the Fracture Strenght of Complete Upper Dentures: A Novel Biomechanical Approach. Eur Sci J. 2019;15:433-48.
- Ravi N, Krishna DP, Manoj S, Chethan A Functional Stress Analysis in the Maxillary Complete Denture Influenced by the Position of Artificial Teeth and Load Levels: an In-vitro Study. J Indian Prosthodont Soc. 2010 Dec;10(4):219-25. doi: 10.1007/s13191-011-0046-0.
- Mankani N, Chowdhary R, Mahoorkar S. Comparison of Stress Dissipation Pattern Underneath Complete Denture with Various Posterior Teeth form: An In Vitro Study. J Indian Prosthodont Soc. 2013; 13(3):212–219. d 10.1007/s13191-012-0193-y.
- Benli, S.; Baş, G. Thermal Stress Analysis of Maxillary Dentures with Different Reinforcement Materials Under Occlusal Load Using Finite Element Method. Appl Sci. 2024; 14: 10271. doi: 3390/app142210271.
- “In the Methodology, regarding the sentence “An impression of the prosthesis was taken for a completely edentulous patient,”it suggests that a patient was involved. Therefore, it would be necessary to indicate approval from an ethics committee and the signing of an informed consent form. No reference to this important aspect is made in the text.”
Answer: Tested prothesis was made as model form old prothesis of unknown patient. Prothesis was used to get an alabaster model for prothesis fabrication. Therefore, test does not include patient and approval from an ethnic committee is not necessary.
- “The workflow is well-structured, but some descriptions (mesh properties, boundary conditions, load application in simulations) require greater detail and standardization to ensure reproducibility. Specify material properties clearly in a table, with sources or testing methods.
Answer: Results of experimental measurements for the PMMA prosthesis acrylic base and a factory-made acrylic prosthesis of a modulus of elasticity and a Poisson's ratio are given in Table 1. Their average values are used for FEM simulations.
Table 1: Mechanical properties of acrylic materials used in FEM simulations [19]
|
Modulus of elasticity E, MPa |
Poisson number n, - |
PMMA acrylic base |
4151 |
0.33 |
PMMA in factory made acrylic tooth |
6315 |
0.30 |
- Figures are informative, but some captions lack sufficient explanation of what is being shown. For instance, in Figures 11–16, include details on scale bars, color meaning, and model orientation. Some figures seem repeated or disorganized, compare the presentation of figure 14 with the rest of the results. Standardize the figures.” About the results and discussion sections, the comparison between eugnathic and progenic relationships is well developed. However, numerical data should be more systematically presented, possibly in tables.”
Answer:
In the FEM simulation, the upper surface of the maxillary prosthesis and the lower surface of the mandibular prosthesis are pressed against the alabaster mold with force. First with 100 N and then with 200 N. When pressed, the force changes into pressure and tension occurs in the model.
We have made the pictures more transparent by giving only one tension table for the entire cliche instead of 4 or 2.
We have described the display in more detail in figures 11 to 16 in the attached discussion.
The fraction of the volume with von MIses stress above 20 MPa in relation to eugnathic and progenic occlusal relationships is shown in Table 2. It is evident that the fraction of the volume with stress equal to or greater than 20 MPa is approximately two times greater in progenic relationships than in eugnathic relationships.
Table 2: The fraction of the volume with von MIses stress above 20 MPa in relation to eugnathic and progenic occlusal relationships
|
Bite force |
Von Mises stress, >20 MPa |
Eugnathic |
100 N |
0.0071% |
|
200N |
0.0352% |
Progenic |
100N |
0.0148% |
|
200N |
0.0710% |
- The discussion could be enriched by referring to the clinical implications in more depth. How these findings affect prosthodontic planning or material selection?
Conclusions are comprehensive but could be more succinct. I suggest to emphasize the clinical and biomechanical implications rather than repeating detailed findings.
Answer: I have made changes to the article. Due to the reviewer's objection that the conclusion was too broad, we have rearranged the chapters. We have moved some of the results to a new Discussion chapter. The Conclusion chapter is also new in a very abbreviated format where I have tried to mention the clinical and biomechanical implications of this research.
The discussion section has been expanded with:
The limit load capacity of complete dentures in eugnathic and progenic occlusal relationships was tested experimentally. We tested the same dentures that we used with alabaster moulds in the eugnathic and progenic occlusal relationship. The test results showed that for full dentures in the eugnathic relationship, the acrylic base between 1left and 1 right and then the acrylic teeth 3 left and 4left cracked longitudinally.
The experimentally determined load curve shows the fit of the acrylic teeth with increasing force of the two complete dentures until the first slip. The first slip leads to a local drop in force, with the first damage occurring in the area of the same compressive force values of 6.14 kN for the eugnathic relation and 7.01 kN compressive force for the progenic relation. Although there were different fracture types of complete dentures in eugnathic and progenic occlusal relations, the maximum bite force values did not differ significantly. At maximum compressive forces, the acrylic base fractures in the eugnathic relationship and the acrylic teeth fracture longitudinally in the progenic occlusal relationship. The cracking of the acrylic teeth or the acrylic base depends on whether the complete dentures are in a eugnathic or prognathic relationship, but in no case did the acrylic teeth detach from the acrylic base. This confirms the hypothesis that the adhesive force of the acrylic base to the acrylic teeth is not a critical point for damage and that special numerical modelling of the adhesive force is not required. In the present study, a numerical simulation of complete dentures in eugnathic and progenic occlusal relationship was performed with two bite forces of 100 N and 200 N, which were taken as reference values for edentulous patients in accordance with the literature [4,13, 18]. The geometry of the complete dentures determined with 3D computed tomography was taken from an edentulous patient whose edentulous ridges were used to fabricate two identical pairs of dentures. The results of the numerical simulations in both occlusal relationships show that there is a redistribution of pressure on the contact surfaces during occlusion.
The results of numerical simulations and other authors Marti-Vigil et al [21] and Kupprano et al [22] showed that the von Mises stresses on the surface reach local values of the same rank with similar mechanical behavior. Prombonas and Dimitris [23] conducted experimental tests of maxillary and mandibular prostheses on hydraulic presses. They applied vertical occlusal forces in the range of 30 to 110 N and measured principal stresses between mandibular and maxillary prostheses in the range of -1.07 to 10.2 MPa. Grachev et al [24] used FEM for computer simulation and prediction of the longevity of removable complete dentures. It was found that both the displacement and the inclination of the individual tooth blocks have a negative influence on the cyclic durability of the dentures.
Ravi et al [25] also applied vertical loads of up to 100 N to test prostheses and measured the applied strains. It was found that a tensile load prevailed in the maxillary denture, while a compressive load prevailed in the posterior palatal region, but the external placement of the maxillary teeth caused a significant decrease in the compressive load.
With a full denture in eugnathic contact, redistribution occurs in such a way that the acrylic base takes over the stresses from the acrylic teeth and, during adaptation, more and more angular dental inserts take over the contact pressures, while the increase in pressure on the incisors decreases. The von Mises equivalent stresses show that the maximum stresses are in the material structure and between the acrylic base and the acrylic teeth. This allows us to find points on the prosthesis where we can expect a crack in the prosthesis or points of damage in the eugnathic occlusal relationship.
With a complete denture in progenic contact, the redistribution is such that the load is transferred to the lateral acrylic teeth and thus the canines and the tips of the incisors and molars remain the most heavily loaded.
The numerical results presented provide an insight into the state of pressure redistribution between the dental contact surfaces and the transfer of load to the acrylic base, thus explaining the relationships that can lead to cracks in the acrylic base or damage to the acrylic teeth.
Although the test results show no significant difference in the static failure strength of total dentures in eugnathic and progenic conditions, the stress behavior of dentures in eugnathic and progenic conditions is different.

Round 2
Reviewer 2 Report
Comments and Suggestions for Authors
Dear Authors.
Thank you for the revisions.
The updated manuscript addresses all previous comments effectively. The abstract and introduction were improved in structure and clarity. References were updated and better contextualized. Ethical concerns were clarified appropriately. Methodological details were enhanced, especially regarding material properties and simulation setup.
Figures were standardized and better described, and the inclusion of numerical tables strengthened the results section. The discussion was expanded with relevant clinical implications, and the conclusion is now more concise and focused.
Finally, the manuscript has improved significantly and is suitable for publication.